# CEP3: Community Event Prediction with Neural Point Process on Graph

**Xuhong Wang**[*]
Shanghai AI Laboratory
wangxuhong@pjlab.org.cn

**Sirui Chen**[*]
University of Hong Kong
ericcsr@connect.hku.hk

**Yixuan He**
University of Oxford
yixuan.he@stats.ox.ac.uk

**Minjie Wang**
Amazon
minjiw@amazon.com

**Quan Gan**[†]
Amazon
quagan@amazon.com

**Junchi Yan**
Shanghai Jiao Tong University
yanjunchi@sjtu.edu.cn

## Abstract

Many real-world applications can be formulated as event forecasting on Continuous Time Dynamic Graphs (CTDGs) where the occurrence of a timed event between two entities is represented as an edge along with its occurrence timestamp. However, many previous works handle the problem in compromised settings, either formulating it as a link prediction task on the graph given the event time, or a time prediction problem for which event will happen next. In this paper, we propose a novel model combining Graph Neural Networks and Marked Temporal Point Process (MTPP) that jointly forecasts multiple link events and their timestamps on communities over a CTDG. Moreover, to scale our model to large graphs, we factorize the joint event prediction problem into three easier conditional probability modeling problems. To evaluate the effectiveness of our model and the rationale behind such a decomposition, we establish a set of benchmarks and evaluation metrics. The experimental results demonstrate the superiority of our model in terms of both accuracy and training efficiency. All the source codes and datasets are available in a GitHub repository.

## 1 Introduction

Modeling dynamic interactions of entities has become an important topic in different applications across many fields [1]. In some cases, some entities, such as those with dense connections, or with similar characteristics, may form certain communities, and communities could also be defined by users based on their criteria. People may only be interested in a specific community of entities' interactions in some practical applications, such as community behavior modeling [2], dynamic community discovery [3] and community outliers detection [4].

We believe that anticipating future occurrences in the community is a task with a high degree of practical significance. For example, tracking the development of social gatherings in a community can help with early epidemic (like COVID-19) intervention [5]. Predicting the behavior of social media communities enables more efficient information recommendation and ad placement.

For better understanding and forecasting the events in a community, we suggest organizing event stream as a Continuous Time Dynamic Graphs (CTDG), and predicting a series of future events not only about **which** two entities will be involved but also **when** they will occur.

CTDG [6] is a common representation paradigm for organizing dynamic interaction event stream over time, with edges and nodes denoting the events with timestamp and the pairwise involved entities,

---

[*]Equal contribution.
[†]Corresponding author.

X. Wang et al., CEP3: Community Event Prediction with Neural Point Process on Graph. *Proceedings of the First Learning on Graphs Conference (LoG 2022)*, PMLR 198, Virtual Event, December 9–12, 2022.

respectively. Formally, denote a CTDG as $G = (V, E_T)$, where $E_T = \{\varepsilon_i : i = 1, \cdots, T\}$ is the set of edges, $\varepsilon_i = (u_i, v_i, t_i)$ with source node $u_i$, destination node $v_i$, and timestamp $t_i$. The edges are ordered by timestamps, i.e., $t_i \leq t_j$ given $1 \leq i \leq j \leq T$. We further denote a CTDG within a temporal window as $G_{i:j} = (V, E_{i:j})$ where $E_{i:j} = \{\varepsilon_k : i \leq k < j\}$. Given the queried community (or node candidates that people are interested in) $C_q \subset V$, predicting $K$ future events within the community given $n$ observed events requires to model the following conditional distribution:

$$p(\varepsilon_{n+1}, \cdots, \varepsilon_{n+K} \mid G_{1:n}, C_q) \tag{1}$$

where the distribution of each edge $\varepsilon_{n+i}$ is further a triple joint probability distribution of its source and destination nodes as well as its timestamp. The community event forecasting task is illustrated using Fig. 1. Compared with traditional time series prediction, event forecasting on a CTDG jointly consider the spatial information characterized by the graph and the temporal signal characterized by the event stream to make a more accurate prediction.

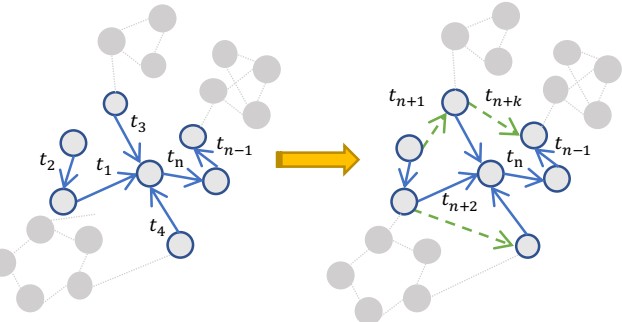

**Figure 1:** Community event forecasting on a CTDG: Given a community (nodes and edges with blue strokes) in a historical CTDG, predict where and when the next interaction event (green arrows) will happen. This process can be repeated to forecast to distant future.

There have been several lines of work to approach the problem but all in compromised settings. Temporal Graph Neural Networks (TGNNs) extends GNNs of static graphs to CTDGs by incorporating temporal signals into the message passing procedure. Most of TGNN progress [7–9] focuses on *temporal link prediction* task, i.e., modeling the conditional distribution $p(v_{n+1} \mid G_{1:n}, u_{n+1}, t_{n+1})$. Other parts of the progress use Temporal Point Process (TPP) for *event timestamp prediction* which predicts the time of the next event but requires the entities to be known, i.e., modeling the conditional distribution $p(t_{n+1} \mid G_{1:n}, u_{n+1}, v_{n+1})$. All these models are not directly applicable to the event forecasting problem on CTDGs.

Marked TPP (MTPP) [10, 11] and its variations such as Recurrent Marked Temporal Point Process (RMTPP) [12] associate each event with a marker and jointly predict the marker as well as the timestamp of future events. MTPP and RMTPP are capable of predicting CTDG events by treating the entity pair $(u_i, v_i)$ as the event marker. However, these kinds of MTPP-based methods face three major drawbacks. To begin with, MTPP methods treat edges as individual makers, which are unable to utilize the community and relationship information, resulting in a suboptimal training solution. Besides, individual makers also bring an $O(|V|^2)$ marker distribution space, making the model less scalable to large graphs. The last difficulty is that RMTPP is further constrained by its recurrent structure, which must process each event sequentially for keeping events' contextual correlation.

As far as we know, Transformer Hawkes [13] is the first progress that incorporates the graph structure information into a temporal point process to jointly predict the incident nodes and timestamp. However, it only supports reasoning future events on the static graph with dynamic node properties, losing the flexibility to process structurally changing dynamic graphics.

Our contributions in this paper are:

**i)** We handle the **C**ommunity **E**vent **P**redicting (**CEP**) task with a graph **P**oint **P**rocess (**CEP3**) model, which is significantly harder than both standalone temporal link prediction and timestamp prediction tasks. Our model incorporates both spatial and temporal signals using GNNs and TPPs and can predict event entities and timestamps simultaneously.

**ii)** To scale to large graphs, we factorize the mark distribution of MTPP and reduce the computational complexity from $O(|V|^2)$ of previous TPP attempts [12, 14] to $O(|V|)$. Moreover, we employ a time-aware attention model to replace the TPP model's recurrent structure, significantly shortening the sequence length of each training step and enabling mini-batches training.

**iii)** We propose new benchmarks for the community event forecasting task on a CTDG. Specifically, we design new evaluation metrics measuring prediction quality of both entities and timestamps. For baselines, we collect and carefully adapt state-of-the-art models from time series prediction, temporal link prediction and timestamp prediction. Our evaluation shows that CEP3 is superior across all four real-world graph datasets. Source code has been already made publicly available.

## 2    Related Work Analysis

### 2.1    Temporal Graph Learning

Temporal Graph Learning aims at learning node embeddings using both structural and temporal signals, which gives rise to a number of works. CTDNE [15] and CAWs [6] incorporate temporal random walks into skip-gram model for capturing temporal motif information in CTDGs. JODIE [7] and TigeCMN [16] adopt recurrent neural networks (RNNs) and attention-based memory module respectively to update node embedding dynamically. Temporal Graph Neural Networks like TGAT [9] and TGN [8] enhance the attention-based message passing process from Graph Neural Networks with Fourier time encoding kernel. These attempts focus on the temporal link prediction task. Besides that, other works [17, 18] focus on the information diffusion task which aims at predicting whether a user will perform an action at time $t$. Nevertheless, none of them is designed for event forecasting.

RE-Net [19] and CoNN [2] study a similar event forecasting setting on Discrete Temporal Dynamic Graphs (DTDGs). However, continuous time prediction is much harder and their methods cannot be directly applied.

### 2.2    Neural Temporal Point Process

A temporal point process (TPP) [20, 21] is a stochastic process modeling the distribution of a sequence of events associated with continuous timestamps $t_1, \cdots, t_n$. The theoretical underpinnings and wide-ranging practical applications of the TPP methods are described in Appendix A. The majority of neural TPP approaches leverage recurrent neural network (RNN) [12, 22–24] based structure to parametrize the stochastic process function and forecast the future events. Such approach cannot be trained in parallel and cannot capture long-term dependencies. Transformer Hawkes [13] replaces the RNN module with a temporal-aware attention transformer to capture temporal dependencies.

However, these methods cannot be directly applied to event forecasting on CTDGs due to the following reasons. First, a CTDG is essentially a *single* event sequence, whose length ranges from tens of thousands to hundreds of millions. RNNs (even LSTMs) are known to have trouble dealing with very long sequences. One may consider dividing the sequence into multiple shorter windows, which will make the events disconnected within the given window and discard all the data before it. This fails to explore the dependencies between events that are distant over time but topologically connected (i.e., sharing either of the incident nodes). One may also consider training an RMTPP with Truncated BPTT [25] on the long sequence as a whole. However, this is inefficient because parallel training is impossible due to its recurrent nature, which means that one has to unroll the sequence one event at a time. Second, although Transformer Hawkes [13] discards the RNN structure, it directly models the marker generation distribution with a unit softmax function, which will produce a vector with space complexity of $O(|V|^2)$, as the event markers will be essentially the events' incident node pairs. This is not applicable to dynamic graphs with changing structure and undesirable for large graphs.

### 2.3    Temporal Point Process on Dynamic Graph

Previous works, such as [14, 26–28], use kinds of recurrent architecture to approximate temporal point process over graphs. However, recurrent architecture prevents the model from parallelized minibatch training, which is undesirable especially on large-scale graphs. This is because learning long-term dependencies using recurrent architecture requires the model to traverse the event sequence one by one instead of using random mini-batch selection.

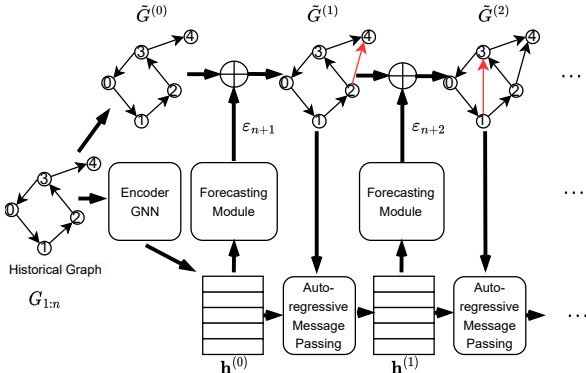

**Figure 2:** The overall architecture of proposed CEP3 model. Red arrows represent the predicted events $\varepsilon_{n+i}$.

MMDNE [29], HTNE [30] and DSPP [31] employ the attention mechanism to avoid the inefficiency of the recurrent structure in training with large CTDGs. However, these works are restrictively dedicated to link prediction or timestamp prediction task. Adapting the two models to event forecasting requires non-trivial changes since neither of them handles efficient joint forecasting of the event's incident nodes.

## 3 Model

Our model is depicted in Fig. 2. To predict the next $K$ events $\varepsilon_{n+1}, \cdots, \varepsilon_{n+K}$ given the history graph $G_{1:n}$, we first obtain an initial representation $\mathbf{h}^{(0)}$ for every node using a GNN Encoder, as well as an initial graph $\tilde{G}^{(0)}$. Then for the $i$-th step, we predict $\varepsilon_{n+i} = (u_{n+i}, v_{n+i}, t_{n+i})$, i.e., the source node, destination node, and timestamp for the next $i$-th event. The predicted event is then added into $\tilde{G}^{(i-1)}$ to form $\tilde{G}^{(i)}$, to keep track of what we have predicted so far. The hidden states $\mathbf{h}^{(i)}$ are then updated from the new graph $\tilde{G}^{(i)}$ and $\mathbf{h}^{(i-1)}$. This generally follows the framework of RMTPP [12], except that **i)** we initialize the beginning states of **Auto-Regressive Message Passing** Module with a time-aware GNN, which allows our recurrent module to traverse over a much shorter sequence without losing historical information; **ii)** we update the **Auto-Regressive Message Passing** network states with a GNN to model the topological dependencies between entities caused by new events; and **iii)** we forecast the nodes and the timestamp for an event by decomposing the joint probability distribution. We give specific details of each component as follows.

### 3.1 Structural and Temporal GNN Encoder

The Encoder GNN Layer should be capable of encoding relational dependencies, timestamps, and optionally edge features at the same time. The encoded node representations are used by the forecaster in next section to generate the predicted events. It has the following form:

$$\mathbf{z}_v^{(l)} = f_{\text{agg}}^{\text{enc}}(\mathbf{z}_v^{(l-1)}, \{g_{\text{msg}}^{\text{enc}}(\mathbf{z}_u^{(l-1)}, \mathbf{e}_{uv}^t, \phi(t_n - t)) : (u, t, \mathbf{e}_{uv}^t) \in \mathcal{N}^v\}) \tag{2}$$

where $\mathcal{N}^v$ is the brief formulation of $\mathcal{N}^v[1:n]$, which represents the subgraph induced by the 1-hop neighborhood of node $i$ on the graph $G_{1:n}$, and $\phi(t)$ are learnable time encodings used in [8, 9, 32]. $f_{\text{agg}}^{\text{enc}}$ and $g_{\text{msg}}^{\text{enc}}$ can be any aggregation and message functions of a GNN-based representation encoder. $\mathbf{z}_v^{(0)}$ could be node $v$'s feature vector, and $\mathbf{e}_{uv}^t$ means the edge feature between node $u$ and $v$ at time $t$.

We use a GNN to initialize the recurrent network's states because it takes the historical events within a topological local neighborhood, including the incident nodes, the timestamps, and the feature of events together as input, while enabling us to train on multiple history graphs in parallel. In particular, we use a neighborhood graph temporal attention based method for encoding, whose detailed formulation is as follows. Temporal Graph Attention Module is a self attention based node

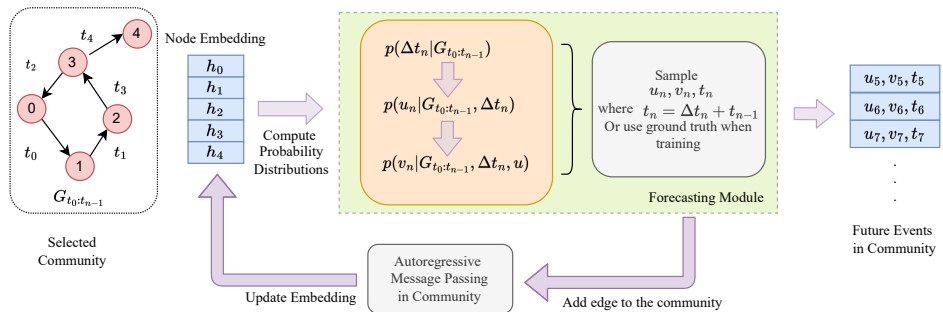

**Figure 3:** The hierarchical probability-chain forecaster and its workflow relationship with the auto-regressive message passing module. The node embeddings are learned from the GNN Encoder described in Section 3.1. Note that the 'selected community' refers to an application-dependent collection of candidate nodes for query.

embedding method inspired by [9], with detailed formulation as:

$$
\begin{aligned}
\mathbf{z}_v^{(l)} &= \texttt{MLP}(\mathbf{z}_v^{(l-1)}||\tilde{\mathbf{z}}_v^{(l)}) \\
\tilde{\mathbf{z}}_v^{(l)} &= \texttt{MultiHeadAttn}^{(l)}(\mathbf{q}_v^{(l)}, \mathbf{K}_v^{(l)}, \mathbf{V}_v^{(l)}) \\
\mathbf{q}_v^{(l)} &= [\mathbf{z}_v^{(l-1)}||\phi(0)] \\
\mathbf{K}_v^{(l)} &= \mathbf{V}_v^{(l)} = \mathbf{C}_v^{(l)} \\
\mathbf{C}_v^{(l)} &= [\mathbf{z}_u^{(l-1)}||\mathbf{e}_{uv}^{t_u}||\phi(t_v - t_u), u \in \mathcal{N}^v]
\end{aligned}
\tag{3}
$$

The multi-head attention with N heads is computed as:

$$
\tilde{\mathbf{z}}_v^{(l)} = \frac{1}{N} \sum_a^{\forall head} \texttt{SoftMax}\left(\frac{(\mathbf{W}_{Q,a}^{(l)}\mathbf{q}_v^{(l)})(\mathbf{W}_{K,a}^{(l)}\mathbf{K}_v^{(l)})}{\sqrt{d_q}}\right)\left(\mathbf{W}_{V,a}^{(l)}\mathbf{V}_v^{(l)}\right)
\tag{4}
$$

where $\mathbf{W}_{Q,a}^{(l)}$ means one head of multi-head attention weight matrix and $d_q$ means the dimension of the vector $\mathbf{q}_v^{(l)}$. The temporal encoding module is the same as in the original papers [8, 9]:

$$
\phi(\Delta t) = \frac{1}{\sqrt{d_w}}\cos(\vec{\mathbf{w}}\Delta t + \vec{\mathbf{b}})
\tag{5}
$$

where $\vec{\mathbf{w}}$ and $\vec{\mathbf{b}}$ are learnable parameters, $d_w$ is the dimension of weight vector $\mathbf{w}$.

Although a GNN cannot consider events outside the neighborhood, we argue that this impact is minimal. We empirically verify this argument by comparing CEP3 against the variant that uses both attention and RNN based memory module in the training phase (named **CEP3 w/ RNN**), which can incorporate historical events and time-aware information by the view of topological locality and recursive impact, respectively. However, RNN takes drastically more memory and time in training because of the same reason in Section 2.3.

### 3.2 Hierarchical Probability-Chain Forecaster

Fig. 3 demonstrates the details of our event forecaster. We can see that the forecaster predicts future events only according to node embeddings and historical connections in the selected candidates (or communities). It means that we do not have to be concerned about a large number of communities which are likely to slow down the process, because our CEP3 model can handle numerous communities simultaneously during training and inference.

From the superposition property [33] of an MTPP described in Section A, we forecast the next event $\varepsilon_{n+i} = (u_{n+i}, v_{n+i}, t_{n+i})$ by a triple probability-chain:

$$
p(u_{n+i}, v_{n+i}, t_{n+i}) = p(t_{n+i})p(u_{n+i} \mid t_{n+i})p(v_{n+i} \mid t_{n+i}, u_{n+i})
\tag{6}
$$

It means that we can predict first the timestamp, then the source node, and finally the destination node. We predict $t_{n+i}$ by modeling the distribution of the time difference as follows:

$$
\begin{aligned}
\eta_v^{(i)} &= \text{Softplus}(\text{MLP}_t(\mathbf{h}_v^{(i-1)})) \\
\lambda_i &= \sum_{v \in V} \eta_v^{(i)} \\
\Delta t_{n+i} &\sim Exponential(\lambda_i) \\
t_{n+i} &= t_n + \Delta t_{n+i}
\end{aligned}
\tag{7}
$$

where $\mathbf{h}_v^{(0)}$ is initialized using the node representation learned by $\mathbf{z}_v^{(L)}$, and Softplus ensures that $\eta_v^{(i)}$ is above zero and the gradient still exists for negative $\eta_v^{(i)}$ values. $\text{MLP}_t$ represents a multilayer perceptron to generate the intensity value of time. Since $\Delta t_{n+i}$ obeys exponential distribution, we simply sample the mean value of time intensity distribution, $\frac{1}{\lambda_i}$, as the final $\Delta t_{n+i}$ output during training. The conditional intensity of the next event occurring on *one* of the nodes in $V$ is thus the sum of all $\eta_v^{(i)}$ [33]. Other conditional intensity choices are also possible.

We then generate the source node $u_{n+i}$ of event $\varepsilon_{n+i}$ from a categorical distribution conditioned on $t_{n+i}$, parameterized by another MLP:

$$
p(u_{n+i} \mid t_{n+i}) = \text{Softmax}(\text{MLP}_{\text{src}}(\mathbf{h}_u^{(i-1)} \| \phi(\Delta t_{n+i})))
\tag{8}
$$

where $\|$ means the concatenation operation and $\phi$ has the same form as in Eq. (2). We then generate the destination node $v_{n+i}$ similarly, conditioned on $t_{n+i}$ and $u_{n+i}$:

$$
p(v_{n+i} \mid t_{n+i}, u_{n+i}) = \text{Softmax}(\text{MLP}_{\text{dst}}(\mathbf{h}_v^{(i-1)} \| \mathbf{h}_{u_{n+i}}^{(i-1)} \| \phi(\Delta t_{n+i})))
\tag{9}
$$

Note that the formulation above will only generate two distributions that have $|V|$ elements, instead of $|V|^2$ as in RMTPP[12]. The implication is that during inference the strategy will be *greedy*: we first pick whatever source node that has the largest probability, then we pick the destination node conditioned on the picked source node. To verify the impact of this design choice, we also explore a variant of our method where we generate a joint distribution of the pair $(u_{n+i}, v_{n+i})$ with $O(|V|^2)$ elements, which we name **CEP3 w/o HRCHY** ('hrchy' is abbreviated for 'hierarchy', the noun of the word 'hierarchical').

The proposed **CEP3** model obeys the probabilistic form of Eq. (6), while the ablation model **CEP3 w/o HRCHY** decomposes $p(u_{n+i}, v_{n+i}, t_{n+i})$ into $p(t_{n+i}) \times p(u_{n+i}, v_{n+i} \mid t_{n+i})$. If a graph has a lot of nodes, evaluating $p(u_{n+i}, v_{n+i})$ will incur a linear projection with $O(|V|^2)$ complexity. This is another obstacle to scaling up to larger datasets.

### 3.3 Auto-Regressive Message Passing

As shown in Fig. 3, we assume that an event's occurrence will directly influence the hidden states of its incident nodes. Moreover, the influence will propagate to other nodes along the links created by historical interactions. Therefore, after generating the new event $\varepsilon_{n+i}$, we would like to update the nodes' hidden states by message passing on the graph with the new events. We achieve this by maintaining another graph $\tilde{G}^{(i)}$ that keeps track of the graph with the historical interactions $G_{1:n}$ and the newly predicted events up to $\varepsilon_{n+i}$.

Specifically, we initialize $\tilde{G}^{(0)}$ with the candidate node set $C$ as its nodes. The resulting graph encompasses the dependency between candidate nodes during the encoding stage. Every time a new event $\varepsilon_{n+i}$ is predicted, we add the event back in: $\tilde{G}^{(i)} = \tilde{G}^{(i-1)} \cup \varepsilon_{n+i}$.

Afterwards, we update the nodes' hidden states using a message passing network such as GCN [34] for spatial propagation and a GRU [35] for temporal propagation:

$$
\begin{aligned}
\mathbf{w}_v^{(i,0)} &= \mathbf{h}_v^{(i-1)} \\
\mathbf{w}_v^{(i,l)} &= f_{\text{agg}}^{\text{upd}}(\{g_{\text{msg}}^{\text{upd}}(\mathbf{w}_u^{(i,l-1)}, \mathbf{w}_v^{(i,l-1)}) : u \in \mathcal{N}_{\tilde{G}^{(i)}}^v\}) \\
\mathbf{h}_v^{(i)} &= \text{GRU}([\mathbf{w}_v^{(i,L)} \| \phi(\Delta t_{n+1})], \mathbf{h}_v^{(i-1)})
\end{aligned}
\tag{10}
$$

where $\mathcal{N}_{\tilde{G}^{(i)}}^{v}$ is the set of the neighboring events of node $v$ in $\tilde{G}^{(i)}$, $f_{\text{agg}}^{\text{upd}}$ can be any message aggregation function and $g_{\text{msg}}^{\text{upd}}$ can be any message function.

To verify the necessity of updating the community using message passing after a event, we also explore a variant where we do not update all the node's hidden states in the community, but only the incident nodes $u_{n+i}$ and $v_{n+i}$. We name this variant **CEP3 w/o AR**.

### 3.4 Loss Function and Prediction

The forecasting module outputs the next event's timestamp $t_{n+i}$ and indicent nodes $u_{n+i}$ and $v_{n+i}$ for all events $\varepsilon_{n+i}$, which are minimized via negative log likelihood. Specifically, the loss function goes as follows:

$$\mathcal{L} = \sum_{i=1}^{K} [\underbrace{-\log(\lambda_i) + \Delta t_{n+i}\lambda_i}_{\text{time loss}} \underbrace{-\log p(u_{n+i}) - \log p(v_{n+i})}_{\text{entity loss}}] \tag{11}$$

where the first two terms within summation are log survival probability from Eq. (14) and the last two terms are log probabilities for source and destination node prediction. The time integration term $\int_{t_n}^{t} \lambda(\tau)d\tau$ in Eq. (14) is approximated using a first order integration method by $\lambda(t)\Delta t$ for the ease of computation.

## 4 Experiments

In this section, we test the performance and efficiency of the proposed methods against several baselines on four public real-world temporal graph datasets: Wikipedia, MOOC [7], GitHub [14], and SocialEvo [36]. The detailed description about datasets is put in the Appendix B. To verify the effectiveness of our CEP3 model, we suggested three ablation models (CEP3 w/ RNN, CEP3 w/o HRCHY. and CEP3 w/o AR) and five advanced methods (GRU, Hawkes Process [37], Poisson Process, RMTPP [12] and Dyrep [14]) as our baseline approach. Almost all baseline methods have been briefly explained in section 2 and 3. In Appendix C we explain why we choose these baselines and provide implementation details for better reproducibility. We also provide the details of our network architecture and the hyper-parameters in Appendix D. The source code is based on PyTorch and Deep Graph Library [38], which is publicly available at `https://github.com/WangXuhongCN/CEP3`.

### 4.1 Evaluation Metrics

For evaluation on a specific dataset, we utilize the communities segmented by the conventional community detection algorithm Louvain [39] as the candidate node set, and we report the average result of all communities.

For each community $C_q$, we measure the perplexity ($PP_{C_q}$) of the ground truth source and destination node sequence for evaluating the node predicting performance, and evaluate the mean absolute error ($MAE_{C_q}$) of the predicted timestamps. Using our MAE to evaluate the quality of auto-regressive forecasting sequence over multiple timesteps can also be seen in traffic flow prediction [40].

Perplexity (PP) [41] is a concept in information theory that assesses how closely a probability model's projected outcome matches the real sample distribution. The less perplexity the situation, the higher the model's prediction confidence. In the field of natural language processing, perplexity is also commonly used to evaluate a language model's quality, i.e., to evaluate how closely the sentences generated by the language model match real human language samples. A language model predicts the next word from the word dictionary, whereas our event predicting model selects nodes from the node candidates (community). Therefore, it is reasonable to employ perplexity as a metric in our task.

Specifically, suppose we have the communities' ground truth event sequence $(u_i, v_i, t_i)$ and the prediction sequence $(\hat{u}_i, \hat{v}_i, \hat{t}_i)$ where $i = 1, \cdots, K$. For each community, we compute perplexity as

$$Perplexity = \exp\left(-\frac{1}{K}\sum_{i=1}^{K}[\log p(u_i) + \log p(v_i \mid u_i)]\right) \tag{12}$$

For the distance between two sequences with difference lengths, we compute MAE [42]:

$$MAE = \frac{1}{K(t_K - t_0)} \sum_{i=1}^{K} \left[ |t_i - \min(t_K, \hat{t}_i)| \right] \tag{13}$$

To keep it comparable in diverse datasets, the MAE is divided by the max time span $t_K - t_0$ and the sequence length $K$. We report the average $PP_{C_q}$ of all communities as the $PP$ of a certain dataset, and $MAE$ is calculated in the same way. Smaller values of both metrics indicate better performance.

## 4.2 Result Analysis

From Table 1 we can see the notable superiority of CEP3 over other baselines in different datasets under both MAE and perplexity. The MAE difference between GRU and RMTPP shows the effectiveness of temporal point process in predicting timestamps. Comparing CEP3 with the sequence-based TPP model RMTPP, we can see that using GNN to capture historical interaction information can improve the forecasting performance. Furthermore, when comparing **DyRep w AR** versus **DyRep** and **CEP3 w/o AR** versus pure **CEP3**, we can conclude that auto-regressive updates can better capture the impact of newly predicted events.

Our model performs better than DyRep w AR because in our CEP3, during the auto-regressive update, the newly predicted event not only influences the node involved in the event but also propagates to other nodes via message passing. It is worth mentioning that our CEP3 model is not only more effective than other baseline models in terms of performance, but it also allows for parallel training and have faster training speed especially in large datasets. The training loss curve and analysis with different parallel sizes are shown in Appendix E.

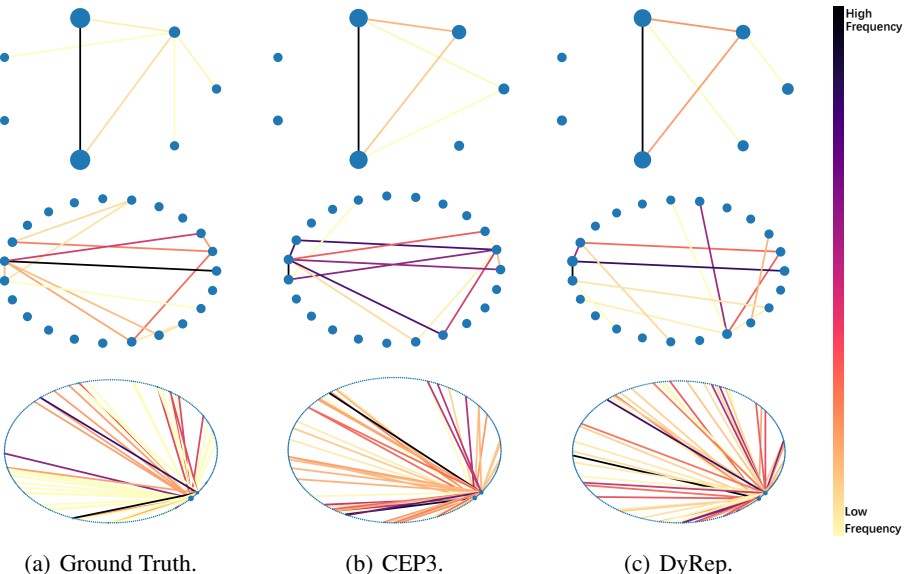

(a) Ground Truth.      (b) CEP3.      (c) DyRep.

**Figure 4:** Prediction visualizations of certain communities in the whole timespan of the test phase. The sizes of plotted nodes indicate their degrees, whereas the colors of edges represent the connection frequencies. Note that the edge colors and node sizes need be compared in the same row. The communities are from GitHub, Wikipedia and MOOC datasets, respectively, from top to bottom.

## 4.3 Forecasting Visualization

From top to bottom, Fig. 4 visualizes the circle layout of a certain community within the graphs of the GitHub, Wikipedia and MOOC datasets, respectively. We plot the ground truth, our model's prediction and DyRep's prediction for comparison. The visualized graphs are generated as follows: We first apply the learned forecasting model to predict the edges using Monte Carlo sampling. This generation process is repeated for three times. We then discard generated edges that are not within

**Table 1:** Comparison of the average and standard deviation of perplexity of incident node prediction and mean absolute error (MAE) of time prediction. The smaller the Perplexity/MAE, the better the performance. The best result is highlighted in **bold** and second best is highlighted with underline. The Rank column shows the average ranking in each metric and dataset (the lower, the better).

| Datasets | Wikipedia | | GitHub | | MOOC | | SocialEvo | | Rank |
|---|---|---|---|---|---|---|---|---|---|
| Metric | Perplexity | MAE | Perplexity | MAE | Perplexity | MAE | Perplexity | MAE | |
| GRU+Gaussian | 131.06 ± 11.27 | 54.54 ± 1.19 | 68.53 ± 1.18 | 59.05 ± 1.72 | 457.40 ± 6.25 | 36.49 ± 2.01 | 33.85 ± 0.27 | 131.71 ± 7.09 | 8.00 |
| Hawkes | 108.00 ± 3.73 | 56.84 ± 0.31 | 74.40 ± 2.47 | 55.21 ± 0.12 | 502.31 ± 12.30 | 36.67 ± 0.29 | 45.33 ± 5.35 | 139.35 ± 0.17 | 9.50 |
| Poisson | 119.19 ± 1.11 | 56.70 ± 0.11 | 61.49 ± 0.96 | 55.21 ± 0.31 | 438.61 ± 7.05 | 36.61 ± 0.78 | 40.48 ± 1.99 | 139.3 ± 1.15 | 8.25 |
| RMTPP w HRCHY | 133.68 ± 2.31 | 34.15 ± 0.89 | 62.19 ± 0.88 | 55.05 ± 1.02 | 616.79 ± 25.74 | 32.29 ± 1.59 | 41.37 ± 6.55 | 140.02 ± 2.06 | 8.88 |
| RMTPP | 121.67 ± 1.01 | 32.91 ± 1.90 | 67.97 ± 1.02 | 54.79 ± 0.47 | 664.07 ± 11.05 | 32.83 ± 2.40 | 37.05 ± 0.77 | 138.9 ± 2.30 | 8.00 |
| DyRep w AR | 116.07 ± 4.98 | 28.74 ± 0.37 | 54.57 ± 1.82 | 28.46 ± 0.65 | 431.18 ± 1.18 | 29.92 ± 1.48 | 29.6 ± 1.93 | 99.96 ± 6.18 | 3.38 |
| DyRep | 119.13 ± 1.02 | 30.04 ± 0.14 | 64.05 ± 0.78 | 36.97 ± 1.74 | 438.61 ± 9.28 | **13.41 ± 1.42** | 36.59 ± 3.02 | 103.01 ± 3.49 | 4.75 |
| CEP3 w/ RNN | 104.87 ± 8.70 | 41.94 ± 1.89 | 60.18 ± 1.04 | 39.22 ± 2.93 | 374.77 ± 24.59 | 20.09 ± 0.33 | 30.37 ± 4.56 | 95.12 ± 2.25 | 3.88 |
| CEP3 w/o HRCHY | **98.98 ± 7.61** | **28.69 ± 0.70** | 52.04 ± 3.33 | **26.8 ± 0.89** | **365.68 ± 28.01** | 31.87 ± 0.18 | **28.66 ± 2.74** | **79.58 ± 5.39** | **1.75** |
| CEP3 w/o AR | 125.51 ± 7.64 | 39.31 ± 2.59 | 61.03 ± 1.03 | 34.03 ± 0.37 | 448.37 ± 4.34 | 21.4 ± 0.47 | 38.59 ± 1.02 | 95.21 ± 4.44 | 6.13 |
| CEP3 | 118.82 ± 4.30 | 32.41 ± 0.58 | **50.42 ± 0.70** | 30.93 ± 1.67 | 401.64 ± 7.06 | 17.69 ± 2.68 | 36.8 ± 1.00 | 94.54 ± 7.31 | 3.25 |

the 33% highest prediction probabilities and obtain the final generated graph. Generating multiple times and then discarding the less possible edges is to reduce uncertainty in each one-time generated graphs.

In the first row, both CEP3 and DyRep capture the triangle connection in this small community. However, the triangle is lighter in the ground truth, which means that DyRep over-reinforces this connection in its predictions. In the second row, the prediction result of CEP3 is more similar to the truth, whereas DyRep generates a high-frequency purple edge which does not exist in the original graph. In the third row, CEP3 successfully learns the two black edges in ground truth, but DyRep predicts more than two links in darker colors.

We can see that our method successfully recognizes the high-degree nodes, and captures many patterns of interactions as well as the evolution dynamics of the interaction graph, which includes those of the nodes with higher degrees. One pressing goal of community event prediction, rather than focusing on a single local node, is to anticipate if a certain high-frequency pattern will emerge in the community from a global perspective. Typical application cases include money laundering patterns [43] in finance and disease transmission patterns [44], etc.

### 4.4 Ablation Study

In Section 3 we have mentioned three variants: **CEP3 w/ RNN** to trade parallelized training for long-term dependency modeling with RNN-based memory module, **CEP3 w/o HRCHY** to compare hierarchical (HRCHY) prediction versus joint prediction of incident nodes, and **CEP3 w/o AR** to compare using an auto-regressive module versus not using one.

**CEP3 w/o HRCHY.** The formulation without hierarchy structure yields a slower training and inference speed as shown in Fig. 5(a). We demonstrate that CEP3 is more efficient on large communities compared to CEP3 w/o HRCHY. As is shown in Fig. 5(a), computing intensity function of each possible node pair would take more time by orders of magnitude. We tackle this issue by decomposing the node pair prediction problem into two node prediction sub-problems, which can be solved quicker especially in large-scale communities.

**CEP3 w/o AR.** The model performance drops significantly and yields similar result as DyRep w AR. From Fig. 5(b), we can see that the AR forecasting is not always useful in all varying numbers of prediction steps. When the number of "prediction steps" is small (such as 10), CEP3 could just use the node embeddings at time $t_n$ to predict the events. When the number of steps becomes larger, the systematic accumulated errors from AR gradually accumulate, leading to low MAE accuracy. And as the number of steps increases, the initial node embedding would have little effect in distant future events. This indicates that without AR, it may be difficult for the CEP3 model to accurately predict long-term occurrences.

**CEP3 w/ RNN.** The results show that the memory module brings performance improvement over pure CEP3, except on the GitHub dataset. The reason is perhaps that the GitHub dataset is a small one with a limited number of nodes and edges, and meanwhile it has a significantly longer timespan than the other datasets. This means that the interaction has low frequency in this situation, causing the memory update to be insufficient.

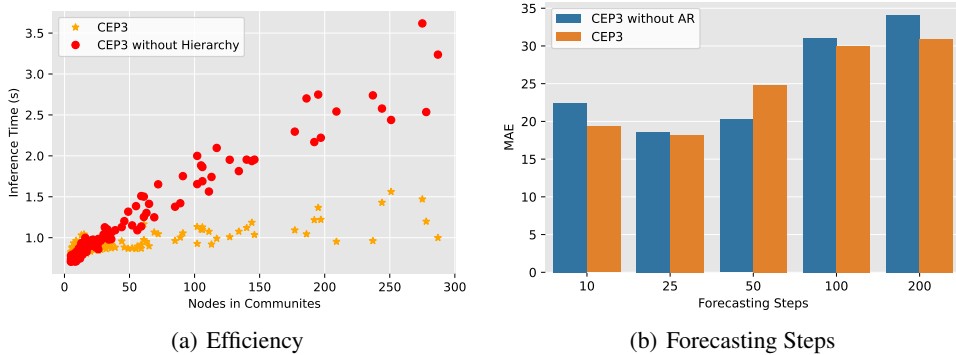

|                |                |
|:--------------:|:--------------:|
| (a) Efficiency | (b) Forecasting Steps |

**Figure 5:** (a) To demonstrate the efficiency of the proposed CEP3 hierarchical probability chain, we show 1000 steps' inference time cost against the node scale. In this scatter figure, each data point represents a community in the Wikipedia dataset (b) The number of forecasting step is an important parameter in almost all forecasting models. To further investigate the effect of the number of forecasting steps and the AR module on the CEP3 model, we run experiments with different numbers of forecasting steps. The smaller the MAE, the better the performance.

## 5 Conclusion and Future Work

In this paper, we have formulated the community event forecasting task on a continuous time dynamic graph and set up benchmarks using adaptations of previous works. We further propose a new model to tackle this problem utilizing graph structures. We also address the scalability issue when formulating the temporal point process on the graph and reduce complexity with a hierarchical formulation. Experimental results show the prediction accuracy and training efficiency of our models.

It is worth mentioning that the concept of community discussed in this study is not limited to communities in traditional network science, it can even be represented with a specific real-world scenario. For instance, monitoring and forecasting the spread of traffic congestion [40] can prevent it from spreading beyond the local area. In a multi-agent system, the community can represent the visible domain of each agent, and the CEP task can be viewed as the prediction of agents' decisions. Consequently, our proposed algorithm has the potential to be implemented in more concrete scenarios, such as autonomous driving, urban simulation, and traffic flow prediction.

However, this work remains two important limitations to solve. One is that we have not provided an experimental evaluation metric of the joint predictions because there have not been any widely accepted metrics that can be used directlyfor now. We plan to consider some metrics from the dynamic graph generation field [45] in future work. The other is that the evaluation is very dependent on the pre-defined communities detected by the Louvain algorithm. Future work will include comparisons utilizing different community detection algorithms (e.g. linkage algorithms [46] and spectral clustering [47]).

## Acknowledgement

This work was partly supported by National Natural Science Foundation of China (62222607, 61972250, 72061127003).

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

# A  Preliminaries of Temporal Point Process

A TPP is mostly characterized by a conditional intensity function $\lambda(t)$, from which it computes the conditional probability of an event occurring between $t$ and $t + dt$ given the history $\{t_i : t_i < t\}$ as $\lambda(t)dt$. According to [48], the log conditional probability density of an event occurring at time $t$ can be formulated as

$$f(t) = \log \lambda(t) - \int_{t_n}^{t} \lambda(\tau)d\tau \tag{14}$$

Additionally, a marked temporal point process (MTPP) associates each event with a *marker* $y_i$ which is often regarded as the *type* of the event. MTPP thus not only models when an event would occur, but also models what type of event it is. Event forecasting over CTDGs can also be modeled as an MTPP if treating the event's incident node pair as its marker. The conditional intensity of the entire MTPP can be modeled as a sum of conditional intensities of each individual marker: $\lambda = \sum_m \lambda_m$. This allows us to first make the prediction of event time, and then predict the marker conditioned on time via sampling from a categorical distribution: $m \sim \text{Categorical}(\lambda_m)$. This avoids modeling time and marker jointly. Such an idea has been widely used in follow-up works such as Recurrent Marked Temporal Point Process (RMTPP) [12]. RMTPP parametrizes the conditional intensity and the marker distribution with a recurrent neural network (RNN). RMTPP's variants include CyanRNN [49] and ARTPP [50].

We also demonstrate several other relevant works and applications related to TPPs [51]. CoEvolving [52], a variant model of MTPP, uses Hawkes processes to model the user-item interaction. NeuralHakwes [22] relaxes the positive influence assumption of the Hawkes process by introducing a self-modulating model. DeepTPP [53] models the event generation problem as a stochastic policy and applies inverse reinforcement learning to efficiently learn the TPP. [54] models link and retweet generation on a social network with a TPP, and also provides a simulation algorithm that generated from the TPP model. It is very similar to our task except that it is focused on a specific social network setting, and the authors did not quantitatively evaluate the quality of the simulation model.

# B  Datasets

Table 2 shows summary statistics of the datasets used in our experiments. A detailed description is put in the below.

**Table 2:** Statistics of the datasets used in our experiments.

| Level | Statistics | Wikipedia | MOOC | GitHub | SocialEvo |
|-------|-----------|-----------|------|--------|-----------|
| Graph level | Edges | 157,474 | 411,749 | 20,726 | 62,009 |
| | Nodes | 9,227 | 7,145 | 282 | 83 |
| | Aver. Event | 34 | 115 | 147 | 1,310 |
| | Aver. Unique Neighbors | 1.98 | 24.98 | 14.65 | 9.02 |
| | Edge Feat. Dim. | 172 | 4 | 10 | 10 |
| | Unique Edges in Graph | 5.99% | 19.95% | 10.27% | 0.62% |
| | Is Bipartite | True | True | False | False |
| | Timespan | 31days | 30days | 1years | 74days |
| | Edges/hour | 211.66 | 576.30 | 2.36 | 7.79 |
| | Data Spilt | 70%-15%-15% by timestamp order | | | |
| Community level | Communities | 142 | 25 | 17 | 10 |
| | Max Nodes | 396 | 990 | 46 | 18 |
| | Aver. Nodes | 50.27 | 264.96 | 15.94 | 7.7 |
| | Max Edges | 4799 | 11686 | 3221 | 12199 |
| | Aver. Edges | 778.28 | 2560.00 | 534.71 | 3420.90 |
| | Min Edges | 77 | 16 | 34 | 863 |
| | Max Edges/hour | 6.47 | 33.33 | 0.36 | 1.99 |
| | Aver. Edges/hour | 1.11 | 5.36 | 0.06 | 0.56 |
| | Min Edges/hour | 0.14 | 0.34 | 0.01 | 0.15 |

**Wikipedia** [7] dataset is widely used in temporal-graph-based recommendation systems. It is a bipartite graph consisting of user nodes, page nodes and edit events as interactions. We convert the text of each editing into a edge feature vector representing their LIWC categories [55].

**MOOC** [7] dataset, collected from a Chinese MOOC learning platform XuetangX, consists of students' actions on MOOC courses, e.g., viewing a video, submitting an answer, etc.

**GitHub** [14] dataset is a social network built from GitHub user activities, where all nodes are real GitHub users and interactions represent user actions to the others' repositories such as Watch and Fork. Note that we do not use the interaction types as we follow the same setting as [14].

**SocialEvo** [14, 36] dataset is a small social network collected by MIT Human Dynamics Lab.

Since the public dataset SocialEvo and GitHub have no edge features, we generate 10-dimensional edge features using attributes including the current degrees of the two incident nodes of an edge, and the time differences between the current timestamp and the last updated timestamps of the two incident nodes. "Percentage of unique edges" in Tab 2 represents the likelihood that new events already happened, and "Average unique neighbors" in Tab 2 measures the likelihood that entities on the graph will seek out connections with other entities that have never interacted with them. Note that the time differences are described in the numbers of days, hours, minutes and seconds, respectively.

## C  Baselines

**Table 3:** Comparison of model capabilities. Note that the usage of RNN prevents a model from parallel training as is mentioned in Section 2.3. *Requires non-trivial adaptation.

| Taxnomy | GNN+TPP | | RNN+TPP | GNN | TPP | |
|---|---|---|---|---|---|---|
| Methods | CEP3 | DyRep | RMTPP | TGAT | Poisson | Hawkes |
| Predicts Link (u,v) | √ | √ | √ | √ | √ | √ |
| Predicts Continuous Time t | √ | √ | √ | | √ | √ |
| Jointly Predicts Event (u,v,t) | √ | √* | √ | | √ | √ |
| Explicitly Models Topological Dependency | √ | √ | | √ | | |
| Complexity of Node Prediction | $O(|V|)$ | $O(|V|^2)$ | $O(|V|^2)$ | $O(|V|^2)$ | $O(|V|^2)$ | $O(|V|^2)$ |
| Parallel Training | √ | | | √ | √ | |
| Captures Sequential Info with | Attention | RNN+Attention | RNN | Attention | Poisson Process | Hawkes Process |

In addition to **CEP3**, **CEP3 w/ RNN**, **CEP3 w/o HRCHY** and **CEP3 w/o AR** mentioned in Section 3, we compare against the following baselines: a Seq2seq model with a **GRU** [35], a Poisson Process (**TPP-Poisson**), a Hawkes Process (**TPP-Hawkes**) [37], **RMTPP** [12] and its variant with the same two-level hierarchical factorization as in Eq. (8) and (9) (**RMTPP w HRCHY**), an adaptation of DyRep [14] and an auto-regressive variant (named **DyRep** and **DyRep w AR**). Notably, RMTPP and its variants are SOTA models for MTPPs in general, and DyRep is SOTA in temporal link prediction and time prediction on CTDG. Since our task is new, we made adaptations to the baselines above, with details of each baseline is as follows:

**Time series Methods**: For baseline models of sequential prediction, we build an RNN model, Gated Recurrent Unit (GRU). Each source and destination cell has a hidden state, the output of the model will be predicted time mean and variance as well as probability for each class to interact, the time will be formulated as Gaussian distribution and source and destination node will be formulated as categorical distribution. This formulation forces GRU predicting timestamp of upcoming events only depending on the hidden state in RNN, whereas other baselines adapt TPP function as a stochastic probability process, obtaining a better modeling capability. We use a **GRU** [35] as a simple baseline without using TPP to model time distribution, treating event forecasting on CTDG as a sequential modeling task. It takes in the event sequence and outputs the next $K$ event in a Seq2seq fashion [56]. The loss term for time prediction is mean squared error and the loss term for source and destination prediction is the negative log likelihood. This formulation forces GRU to predict the timestamp of upcoming events only depending on the hidden state in RNN, whereas other baselines adapt TPP for better modeling capabilities.

**Temporal Point Process Methods**: Following the benchmark setting of [12], we compared our model against other traditional TPP models and deep TPP models.

**Table 4:** Configurations for our CEP3 and all baselines.

| Name | Value |
| --- | --- |
| Hidden Dim in Encoder | 100 |
| Hidden Dim in Forecaster | 50 |
| Hidden Dim in Time Encoding | 100 |
| Layers in MLPs | 2 |
| K-hops | 2 |
| Sampled neighbors/Hop | 15 |
| Learning rate | 0.0001 |
| Optimizer | Adam |
| # of attn head | 4 |
| Recurrent Module | GRU |
| Epochs | 100 |
| Forecasting Window | 200 Steps |
| Community Detection Method | Louvain |

- **TPP-Poisson**: We assume that the events occurring at each node pair $(u, v)$ follows a Poisson Process with a constant intensity value $\lambda_{u,v}$, which are learnt from data via Maximum Likelihood Estimation (MLE).

- **TPP-Hawkes** [37]: We assume that the events occurring at each node pair $(u, v)$ follows a Hawkes Process with a base intensity value $\mu_{u,v}$ and an excite parameter $\alpha_{u,v}$, which are learnt from data via MLE.

- **RMTPP** [12]: We directly consider each source and destination node pair as a unique marker. We note that this formulation will exhaust memory and time on graphs with more than a few thousand nodes, since RMTPP will assign a learnable embedding for each node pair, resulting in $O(|V|^2)$ (here $|V|$ is total number of nodes in the entire CTDG) parameters which is too expensive to update.

- **RMTPP w HRCHY**: We consider a variant of RMTPP where we replace the source-destination node prediction with our hierarchical formulation: we first select the source node, then condition on the source node we select the destination node. The latter formulation can also serves as an ablation study to demonstrate the usage of considering the graph structure.

- **DyRep** [14]: DyRep is a popular work that combines the temporal point process with graph learning techniques to model both temporal and spatial dependencies. Since the original DyRep formulation only handles temporal link prediction and time prediction, but not autoregressive forecasting, we compute an intensity value $\lambda_{u,v}$ with DyRep for each node pair and assume a Poisson Process afterwards.

- **DyRep w AR**: We make a trivial adaptation to the original formulation of DyRep by updating the source and destination node involved once a new edge is added to the graph. The update function is identical to DyRep's updating function during embedding. This benchmark is designed to demonstrate that the propagation from newly forecast event to local neighborhood is necessary in getting better performance.

We also crave our proposed model for having the ability to implement minibatch training. As described in the beginning of Section 3, our CEP3 achieves large-scale and parallelized training by utilizing Hierarchical TPP and GNN-based updating module, respectively. A summary of the mentioned baselines is shown in Table 3, where we conclude that the proposed model CEP3 satisfies all the desirable properties.

# D  Configuration

In this section, we provide the details of our network architecture, the hyper-parameters and the selected community detection method for better reproducibility. Table 4 summarizes key parameters in our model. For all the experiments we train our model and benchmark models on Intel(R) Xeon(R) Platinum 8375C CPU @ 2.90GHz.

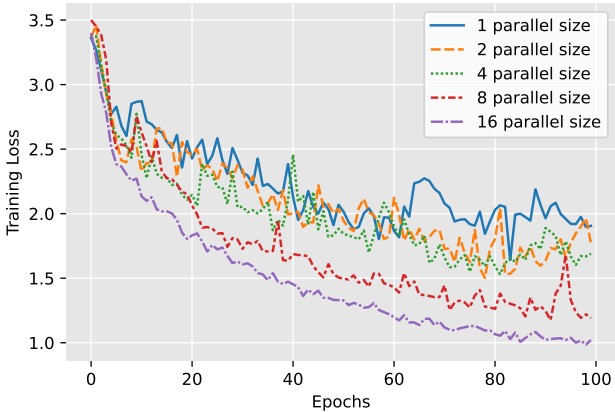

**Figure 6:** Training loss curve with different parallel sizes.

# E   Parallel Training

Inspired by the Transformer [57] models in NLP, we believe it is essential to use a pure attention-based model in temporal graph encoders. This is because using a pure attention-based GNN as an encoder enables us to train multiple time windows in minibatches as described in Section 3.1, whereas models such as Dyrep and RMTPP cannot utilize parallel training due to their RNN structures. This property allows our model to benefit from mini-batch training such as gradient stabilization and faster convergence. Fig. 6 shows our experiment result on the Wikipedia dataset with different numbers of parallel processes, and suggests that using parallel training can increase the speed significantly without losing accuracy.

