# OpenReview forum: "CEP3: Community Event Prediction with Neural Point Process on Graph"
_logconference.io/LOG/2022/Conference — LoG 2022 Poster_

### Official Review · Reviewer_cWCF · 2022-10-17

**Overall Score:** 6
**Confidence:** 3

**Review:**

##########################################################################

Summary:

The paper looks at the task of event prediction within communities of Continuous Temporal Dynamic Graph (CTDG). It aims at jointly predicting the event time and the two nodes involved in the event with the CEP3 method. CEP3 combines a GNN encoder, a MTPP forecaster, and a Auto-regressive message passing component to break the joint probability on event type and event time in conditional probabilities which is more scalable w.r.t. the number of nodes. The paper also propose evaluation experiments to measure the quality of entities and timestamps predicitons.

##########################################################################

Reasons for score:

Overall, I vote for weak reject. The event prediction is clearly introduced and well formalized. My
major concerns are about the model presentation and the experimental setup (see cons below). Hopefully the authors can address my concern in the rebuttal period.

##########################################################################

Pros:

1. The presentation of the event prediction task on CTDG communities is clear and mathematically well formalized.
2. The CEP3 model combines different techniques to solve a new task in a fairly scalable fashion.
3. This paper provides experiments which evaluate different parts CEP3. It includes an ablation study and the evaluation of both entities and event time predictions.

##########################################################################

Cons:

1. Related Work: The related work description is spread over the paper and the appendix. This makes sometimes redundant or harder to identify relevant related works. More specifically l. 50-71 sounds a bit redundant with e.g. section 2.1. Further, the TPP related works cite only two related works while the literature is quite rich in this field as described in [28]. Some related works for TPP are only mentioned in appendix. Action suggestion: I feel that concentrating the related work description at one place would improve the paper. I would also extend the TPP related works by e.g. using the survey [28] and partly moving appendix B to the main paper.
2. Model:
    1. The model description (sec.3) is sometimes hard to follow. The paper introduced a very large number of mathematical notations. Eq.(3)would need some explanation even if it relies on previous works. What is the meaning of each variable in Eq.(3) ? The meaning of bold variables was unclear to me. Should bold variables be used for all vectors/matrices ? What is the difference between the bold and not bold $z_i^{(l), (t)}$ in (3)? Should the vectors be denoted with arrows like in Eq.(5)? The notations are sometimes not consistent (e.g. the neighorhood of $v$ in Eq.(2) and l. 147, probably a typo). Action suggestion: Only present necessary equations in the main text to reduce the number of notations. Make mathematical notations more consistent.
    2. It is not clear to me why forecaster is “Hierarchical” and it is not explained in sec. 3.2. Action suggestion: Explain the "Hierarchy" aspect.
    3. “Specifically, we initialize $ \hat{G}_0$ with the candidate node set C as its nodes. Two candidate nodes are connected in $ \hat{G}_0$ if their distance is within L hops. The resulting graph encompasses the dependency between candidate nodes during the encoding stage” .This sentence was unclear to me.
3. Experiments:
    1. The evaluation is very dependent of the pre-defined communities. The communities are computed with only the Louvain algorithm which heavily suffers from the resolution limit especially for large graphs (Resolution limit in community detection, Fortunato et al.). It would be interesting to report results for different community sizes and number of communities for each dataset. Action suggestion: Take other clustering algorithms (e.g. linkage algorithms, spectral clustering) to define communities in the experiments. Performs the experiments when the number of communities changes. This is also possible with the Louvain algorithm by changing the resolution parameter.
    2. Eq. (12) is supposed to evaluate the predictions (ˆui,ˆvi,ˆti) but these predictions notations do not appear in the PP formula.
    3. Eq.13 compares t_i and ^t_i while the true even might be different from the predicted event. Thus, it is possible to achieve good MAE while the model is completely wrong in terms of entities. Since a key contribution is the joint prediction, it would be more convincing to provide an experimental evaluation of the joint predictions. Action suggestion: Explicitly mention the limitations of this evaluation. Propose an experimental evaluation of the joint predictions.
    4. I appreciate the will to show results visualization as in Fig. 3. However, Fig. 3 did not convince me that CEP3 is better than DyRep in this specific case. DyRep does not less similar to the ground-truth than CEP3. Action suggestion: Maybe another color scheme would show a better visualization. Another idea is to complement the plots with a quantitative metric measuring the distance to the GT next to each plot.

Others:

- I feel that it would be appropriate to cite the work(s) who introduced CTDG framework in line 30. Without these citations, it is hard to understand where this common representation comes from.
- typo "the/a" l.82

I m happy to improve my score if a majority of the above points are addresses (e.g. with action suggestions).

#### Post Rebuttal

I believe that authors improved the paper by providing clarifications and discussing the limitations of the work. Therefore, I increased my score from 5 to 6.

---

### Official Review · Reviewer_EuMf · 2022-10-19

**Overall Score:** 6
**Confidence:** 3

**Review:**

This paper mainly studies the forecasting problem on continuous-time dynamic graphs. The main motivation is to jointly forecast multiple link events and their timestamps over dynamic graphs. For this aim, the authors propose a united model composed of graph neural networks and marked temporal point process. For scalability, the authors further propose to factorize the joint prediction problem into three easier conditional probability modeling problems. Experiments are conducted to show the improved performance in effectiveness and efficiency.

Strengths:
1. The paper is easy to read; the organization of this paper is clear.
2. It is well-motivated for the studied problem. It's interesting to jointly consider forecasting link events and timestamps on dynamic graphs.
3. The experiments part seems to be convincing with the new benchmark for the community event forecasting task.

Weakness:
1. It is an incremental work of existing forecasting methods on dynamic graphs.
2. In the method part, some proposed architectures are not explained very well. For example, why design hierarchical probability-chain architectures as forecaster? Is it better for performance? It will be better if some intuitions are given.
3. In the experiments part, some recent baseline algorithms are missing for comparison.

---

### Official Review · Reviewer_XRhC · 2022-10-19

**Overall Score:** 3
**Confidence:** 4

**Review:**

The paper describes a model for predicting events in a dynamic graph. Unlike most previous work, the model predicts both the incident nodes and the time step of the event jointly, rather than only predicting one given the other. The model formulation and training is based on temporal point processes.

* Suggestions:
	1. The paper essentially proposes a new task, a new model, new baselines, and a new evaluation metric, which makes it difficult for the reader to judge how effective the model actually is. I think carefully designed, easy to interpret baselines are therefore crucial. Currently the authors only compare against neural network baselines, which are all fairly similar to the proposed model. I would strongly suggest comparing to a more straightforward baseline, for example simply predicting past events with the same time interval as before (or an average time interval). In my experience, dynamic graph data sets largely consist of repeated events meaning even such a simplistic baseline might perform fairly well. The examples in Figure 3 are encouraging but it is hard to judge how representative they are.
	2. In a similar vein as my comment above, it would be informative to add the percentage of unique edges to the data set statistics and compare it to the predictions of the model(s). Again, this would provide more context for the results.
	3. Section 3, in particular subsection 3.1 could benefit from some major improvements in terms of clarity. The paper claims that the model does not require unrolling because it uses a pure attention mechanism. However, at the same time, it makes repeated references to recurrent states (eg. l. 139-140, 151 among others). It is unclear whether this refers specifically to the version of the model with an RNN (“CEP w RNN”) or whether it is common to all models. Furthermore, subsection 3.3 refers to auto-regressive message passing, which seems to also require a hidden state. A more structured, clear exposition would be helpful.

Minor comments:
* “Exponential” in Eq. 7 is typeset incorrectly

I can see a number of positive aspects about this work:
* The task is carefully and elegantly designed and appears more practically useful than the formulations addressed in prior work. I believe having an effective, well-motivated model for this task would be a great addition to the literature in this field and the proposed model looks like a step in the right direction.
* It is commendable that the authors consider the scalability of their approach. In my experience, a lot of prior work is computationally expensive and this task is particularly interesting for large, production-grade graphs.

While the paper primarily combines existing neural network components I believe the novelty of the work is sufficient for publication given it addresses a relevant task that is of interest to the wider community.

However, in its current form I am reluctant to recommend acceptance simply because the merit of the paper hinges a lot on the effectiveness of the proposed model, which to me remains unclear. If the authors could incorporate some of my suggestions, above all an additional, easy to interpret baselines, I am happy to raise my score.

---

### Official Review · Reviewer_Pg5E · 2022-10-20

**Overall Score:** 6
**Confidence:** 4

**Review:**

The paper presents a method to predict when and which two nodes in a graph community will be linked (i.e., when and what event will happen). Rather than taking the whole graph into account, the paper first leverages community detection algorithms to divide the network into subgroups and then performs event prediction within each group. More specifically, GCN or message passing is utilized to capture the topological information and the temporal point process is utilized to capture the temporal information.

Strong points:
1. The paper studies an interesting problem, it is of practical significance to predict events within a community of a graph.
2. The paper is well-motivated and shows clearly the difference between it nad existing methods.
3. It is not hard to understand the main idea of this work and the proposed model, albeit simple, indeed makes sense.
4. The paper did lots of ablation studies to verify the effectiveness of the proposed model.
5. Code enclosed.

Weak points:
1. The paper repeatedly claims that it is the first to jointly predict the next event's incident nodes and timestamp within a certain community. However, existing methods can already achieve this goal (e.g., Transformer Hawkes process and its follow-up work [1]).
2. Although the main idea of this work is clear, the description of the model (Sec. 3) is a bit hard to follow due to the huge amount of notations used. Moreover, many symbols are not clearly explained, some notations are in bold and some are not and the subscripts and superscripts are also confusing (e.g., Eqs (2) and (3)). Is Eq. (5) correct?
3. The hierarchical probability-chain forecaster is straightforward and I cannot learn too much from this design. Besides, one concern is that in practice some links are undirected, in such cases will the order of the factorized terms matter?
4. In Table 3, it is shown that the proposed model underperforms CEP3 without using hierarchical factorization significantly. Although CEP3 is faster, the big drop in performance makes the model less attractive.
5. Stronger baselines (e.g. Transformer Hawkes Process) and effects of key parameters (e.g., L) should be included. It is also super important to use different community detection algorithms to divide the network into subgroups.

Questions:
1. Why is AR forecasting not useful sometimes? It looks like that using AR forecasting should do no harm to the model performance.

[1] Zuo, Simiao, et al. "Transformer hawkes process." International conference on machine learning. PMLR, 2020.

---

### Meta-Review · Area_Chair_w74b · 2022-11-17

**Confidence:** 4
**Recommendation:** Accept

**Meta Review:**

This paper studies predicting the evolution of dynamic graphs.  While there's a substantial body of work already in this area, reviewers appreciated the paper's motivation and the joint problem setting of predicting both links and time steps.

Strengths
+ S1. Experimental results show lift for the proposed method.
+ S2. The ablation study provides insight into the problem.

Weaknesses
- W1. Some reviewers thought the result was incremental
- W2. Evaluation based on a new benchmark, which is generally a red flag.

While reviews are 'lukewarm', I think the discussion with reviewers was productive and improved a lot of issues with the paper.  In my opinion its basically ready for acceptance now.   I encourage the authors to provide both additional experimental results, and to do a thorough read through for the final version.

---

### Decision · Program_Chairs · 2022-11-22

**Decision:**

Accept (Poster)

**Comment:**

We agree with the AC that this paper is ready for publication. We encourage authors to incorporate suggestions for clarity improvements, in particular those mentioned by reviewer `XRhC`.